# Artificial intelligence's contribution to biomedical literature search: revolutionizing or complicating?

Rui Yip[1,2], Young Joo Sun[1,2], Alexander G. Bassuk[3], Vinit B. Mahajan[1,2,4]*

1 Molecular Surgery Laboratory, Stanford University, Palo Alto, California, United States of America, 2 Department of Ophthalmology, Byers Eye Institute, Stanford University, Palo Alto, California, United States of America, 3 Department of Pediatrics, University of Iowa, Iowa City, Iowa, United States of America, 4 Veterans Affairs Palo Alto Health Care System, Palo Alto, California, United States of America

* vinit.mahajan@stanford.edu

## Abstract

There is a growing number of articles about conversational AI (i.e., ChatGPT) for generating scientific literature reviews and summaries. Yet, comparative evidence lags its wide adoption by many clinicians and researchers. We explored ChatGPT's utility for literature search from an end-user perspective through the lens of clinicians and biomedical researchers. We quantitatively compared basic versions of ChatGPT's utility against conventional search methods such as Google and PubMed. We further tested whether ChatGPT user-support tools (i.e., plugins, web-browsing function, prompt-engineering, and custom-GPTs) could improve its response across four common and practical literature search scenarios: (1) high-interest topics with an abundance of information, (2) niche topics with limited information, (3) scientific hypothesis generation, and (4) for newly emerging clinical practices questions. Our results demonstrated that basic ChatGPT functions had limitations in consistency, accuracy, and relevancy. User-support tools showed improvements, but the limitations persisted. Interestingly, each literature search scenario posed different challenges: an abundance of secondary information sources in high interest topics, and uncompelling literatures for new/niche topics. This study tested practical examples highlighting both the potential and the pitfalls of integrating conversational AI into literature search processes, and underscores the necessity for rigorous comparative assessments of AI tools in scientific research.

## Author summary

As generative Artificial Intelligence (AI) tools become increasingly functional, the promise of this technology is creating a wave of excitement and anticipation around the globe including the wider scientific and biomedical community.

purpose. The work is made available under the Creative Commons CC0 public domain dedication.

**Data availability statement:** Data can be accessed through the Supplementary Information file.

**Funding:** VBM is supported by NIH grants (R01EY031952, R01EY031360, R01EY030151, and P30EY026877), the Stanford Center for Optic Disc Drusen, and Research to Prevent Blindness, New York, New York. AGB is supported by NIH grants R01EY030151 and R01EY031952. The funders had no role in study design, data collection and analysis, decision to publish, or preparation of the manuscript.

**Competing interests:** The authors have declared that no competing interests exist.

Despite this growing excitement, researchers seeking robust, reliable, reproducible, and peer-reviewed findings have raised concerns about AI's current limitations, particularly in spreading and promoting misinformation. This emphasizes the need for continued discussions on how to appropriately employ AI to streamline the current research practices. We, as members of the scientific community and also end-users of conversational AI tools, seek to explore practical incorporations of AI for streamlining research practices. Here, we probed text-based research tasks—scientific literature mining—can be outsourced to ChatGPT and to what extent human adjudication might be necessary. We tested different models of ChatGPT as well as augmentations such as plugins and custom GPT under different contexts of biomedical literature searching. Our results show that though at present, ChatGPT does not meet the level of reliability needed for it to be widely adopted for scientific literature searching. However, as conversational AI tools rapidly advance (a trend highlighted by the development of augmentations in this article), we envision a time when ChatGPT can become a great time saver for literature searches and make scientific information easily accessible.

## Introduction

Artificial intelligence (AI), in its many forms, has been heavily incorporated into scientific research, bringing significant benefits in big data analysis and automation of routine tasks [1–6]. Among these advancements, large language models (LLMs) and its application as a conversational AI (processing and generating human-like, conversational text; e.g., ChatGPT) enable those without AI expertise to easily leverage its capabilities [7]. The rapid developments of LLMs have driven members of the scientific society to widely investigate its application in the biomedical field, which have been shown to be effective in diagnostics, medical education, and even in gene-editing protein designs [8–12].

 Scientific information mining is one of the fundamental steps of the scientific discovery process. It involves a combined process of literature search followed by a subjective evaluation to identify good quality references on a specific topic [13]. However, this is a laborious and time-consuming task. Surveys indicate that scientists spend about 7 hours weekly on literature searching, while a literature review on a particular topic takes an average of 41 weeks for a five-person research team [14]. Due to its capability of generating an immediate response, ChatGPT has been investigated if it can be effectively utilized to streamline this process, by asking it to write and synthesize literature reviews and summaries. Yet, the findings showed limitations, with persistent issues in—accuracy (commonly known as hallucination, when ChatGPT fabricated information), consistency (when ChatGPT showed different responses each time to the same query), and relevance (when ChatGPT returned irrelevant information). These limitations can be critical especially in a literature review context, as biomedical researchers strictly require robust, reproducible,

and accurate information/findings. Many have, therefore, concluded that ChatGPT currently, may not be a robust tool for generating research reviews [15–22].

We evaluated ChatGPT's performance in a more straightforward task - its ability to identify high quality references under various timely biomedical literature search scenarios. In doing, so we identified ChatGPT's potential and its current limitations when used for biomedical literature searching.

## Results and discussions

### Strategies of employing ChatGPT for scientific literature search

**Conventional approaches in scientific literature searching.** Literature searches conventionally rely on web-based search engines and scientific literature archives/databases (i.e., Google or PubMed) and generally follow the steps below (**Fig 1A**):

1. **Keyword Formulation**: List essential keywords representing your research focus.

2. **Search Execution**: Input keywords into search engines or databases.

3. **Initial Screening**: Skim through the returned results, assessing titles and abstracts for relevance.

4. **Deep Dive**: Read relevant papers in detail to understand the author's findings, extract information, evaluate and accommodate author's findings/arguments/discussions/opinions.

5. **Reference Mining**: Examine cited works in selected papers to uncover further relevant reads that offer additional/contrasting insights or information that is not present in the initial studies.

6. **Iterative Review**: Continue the cycle of reading and reference mining until either no new relevant information is identified, or enough information is gathered/acquired.

Using conventional web-based search engines involves time-consuming steps (3–6), as users must sift through excessive information and discard what is irrelevant. This challenge intensifies for scientists lacking prior knowledge in the field, topic, or technologies being searched. Conversational AI, like ChatGPT potentially offers a more time-efficient way by automating the search, extraction and analysis of information (steps 2–4), providing immediate response to specific inquiries like "Give me 6 vitreous proteomics papers in age-related macular degeneration (AMD)". To investigate this, we tested ChatGPT for its efficacy in identifying vitreous proteomics studies in AMD. Throughout this process, we continuously proofread the results and provided a quantitative analysis to assess its performance.

Before AI tools, literature searches began by asking web-based search engines (i.e., Google, PubMed, and Google Scholar) to look for "Vitreous proteomics studies in AMD". After a series of exhaustive literature searches using traditional platforms, we identified 10 papers that were relevant. Google returned 116,000 search results, with the top 10 results, identifying six relevant papers (**Fig 1B**). Querying the same prompt with PubMed, six results were returned, and only one was relevant. It is notable that PubMed offers functions for optimizing search results, such as the use of Boolean operators or filter results by journal categories and study types (**Fig 1C**). For this study, we did not employ optimization strategies, as our goal was to compare search outcomes from the perspective of an average end-user in typical research scenarios. Google Scholar, another popular academic search platform, produced 4,460 results for the same query. Upon proofreading the top ten results, four suggested links correctly identified relevant papers (**Fig 1D**).

**Scientific searches with basic functions of ChatGPT.** Employing the same inquiry, "Give me 6 vitreous proteomics studies in AMD," we tested both ChatGPT's GPT-3.5 and ChatGPT Classic (GPT-4 with no additional capabilities) models (**Fig 1E**). Concerns from the scientific community regarding ChatGPT include inconsistent outputs, inaccurate or fabricated references, and inclusion of irrelevant articles [16,23,24]. To address this, we prompted the same question 10 times and analyzed the concerns for basic versions of ChatGPT (i.e., GPT-3.5 and ChatGPT Classic).

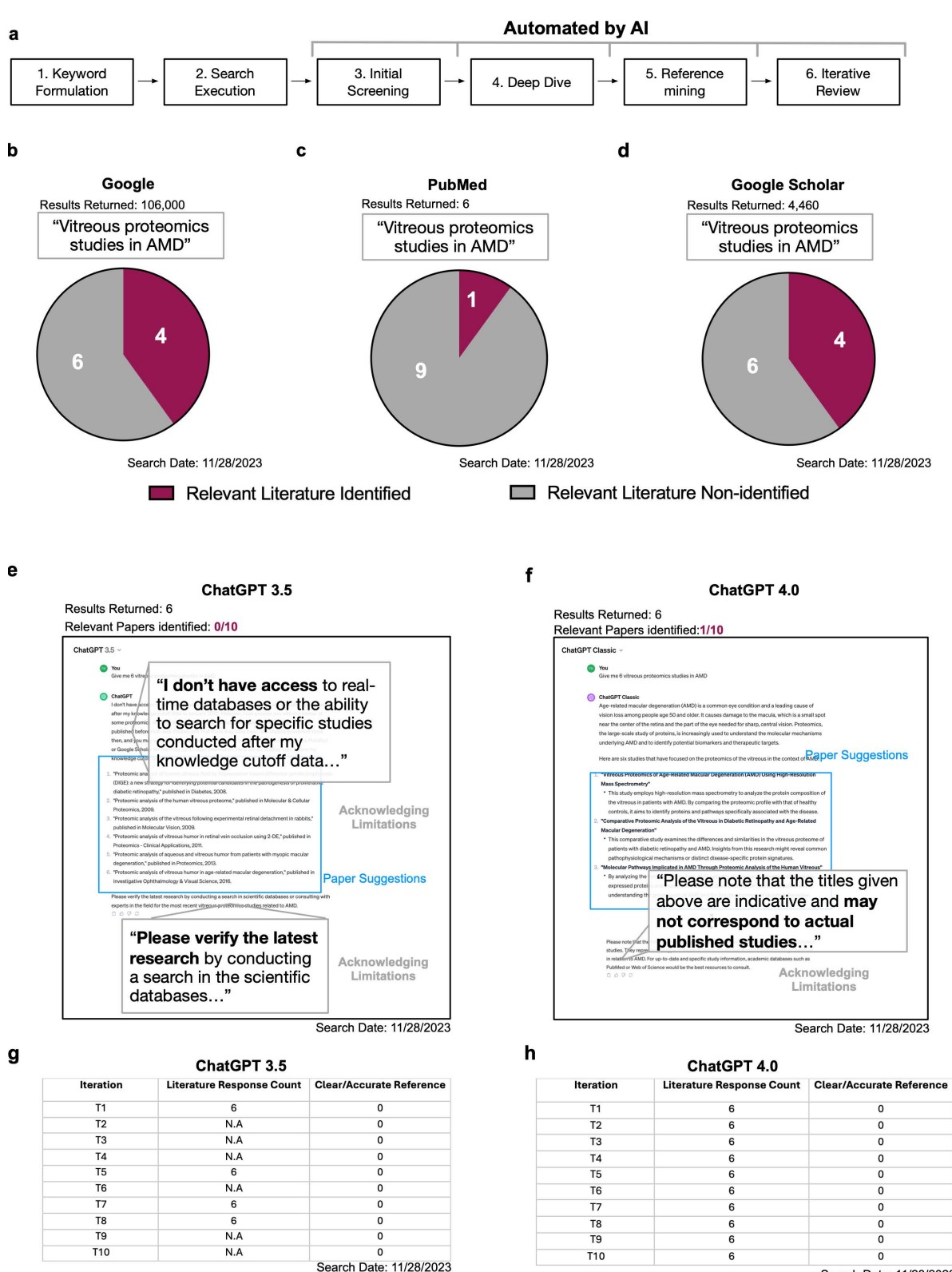

**Fig 1. Can scientific literatures search be aided by conversational AIs.**

GPT-3.5 generated inconsistent results, failing to suggest relevant publications in six instances and providing inaccurate references in others, often advising users to consult PubMed and Google Scholar. In contrast, while ChatGPT Classic generated lists of publications in every iteration, it also failed to provide accurate references, often featuring rephrased words in titles, fabricated authorships, or incorrect publication dates or journals (**Fig 1F**). As such, based on their basic functions alone, neither GPT-3.5 nor ChatGPT Classic demonstrated a clear advantage over conventional search methods with clear limitations in consistency, accuracy, and relevancy that could lead to scientific misinformation.

**Scientific literature searching with augmented ChatGPT.** The use of conversational AI involves 3 elements: the AI model (Large language model; LLM), the data source, and the user input. Augmentations to each can enhance ChatGPT's output quality—through user-support tools/functions and prompt engineering (https://platform.openai.com/docs/guides/prompt-engineering) [25,26]. To assess if these augmentations could improve ChatGPT's consistency, accuracy, and relevancy, we tested them in an analogic framework—the use of conversational AI seen as "a student going into a library to ask a specific question." Here, the student is the user, and the librarian is the LLM providing answers based on the information available in the library (the data source). Below shows our augmentation approach for each element (**Fig 2**):

1. **Expansion of library**: use of online and/or plugin data source.

   • Access to real-time web through ChatGPT's web-browsing function.

2. **The change to specialized librarian**: use of plugins specialized in scientific literature search.• Tailoring how the large language model interprets and conveys information with the "Scholarly" plugin.

3. **Asking better questions**: prompt engineering.

   • Refining the input to enhance the quality and relevance of responses from LLM.

Using the same prompt, "Give me 6 vitreous proteomics studies in AMD," we tested it in ten separate iterations for each augmentation strategy. Post the November 6, 2023 update, ChatGPT4's default model included a built-in web-browsing function. In five of ten iterations, at least one relevant study was identified. Surprisingly, there was an iteration with all six answers having clear and accurate references and five being relevant, including a study that we previously did not identify. The incorporation of web-browsing functions enabled the LLM to access more current information available on the internet and improved search responses. However, issues with inconsistency and accuracy persisted. For instance, one hyperlink erroneously led to a paper on sugar tolerance instead of vitreous proteomics (**Fig 2A**).

We employed prompt engineering in the multimodal ChatGPT-4 model that included a more detailed prompt with clearer instructions that encompasses the purpose of the inquiry ("conducting a literature search"), specified the information source ("peer-reviewed articles from academic and scientific journals"), and the detailed study designs sought ("… patients with AMD" and "included mass spectrometry and/or multiplex ELISA data") (**Fig 2B**). With these refinements, ChatGPT demonstrated an enhanced ability to distinguish relevant from irrelevant studies. These prompt refinements align with the prompt engineering strategy of providing clear instructions, as recommended by OpenAI. A clearer and more detailed prompt improves the LLM's response by narrowing the scope and reducing errors caused by ambiguity. Additionally, the extra information acts as a filter, guiding the LLM to exclude studies that do not meet the additional criteria (https://platform.openai.com/docs/guides/prompt-engineering). However, it is important to note that challenges with inconsistency and accuracy continued.

Next, we integrated a plugin for searching academic and journal-published articles (i.e., Scholarly) (**Fig 2C**). In ten iterations, it consistently identified six papers with accurate references, repeatedly returning the same four relevant papers. This improvement stems from Scholarly's ability to access specialised academic repositories and its built-in constraints, ensuring the LLM to only retrieve data exclusively from these sources [27].

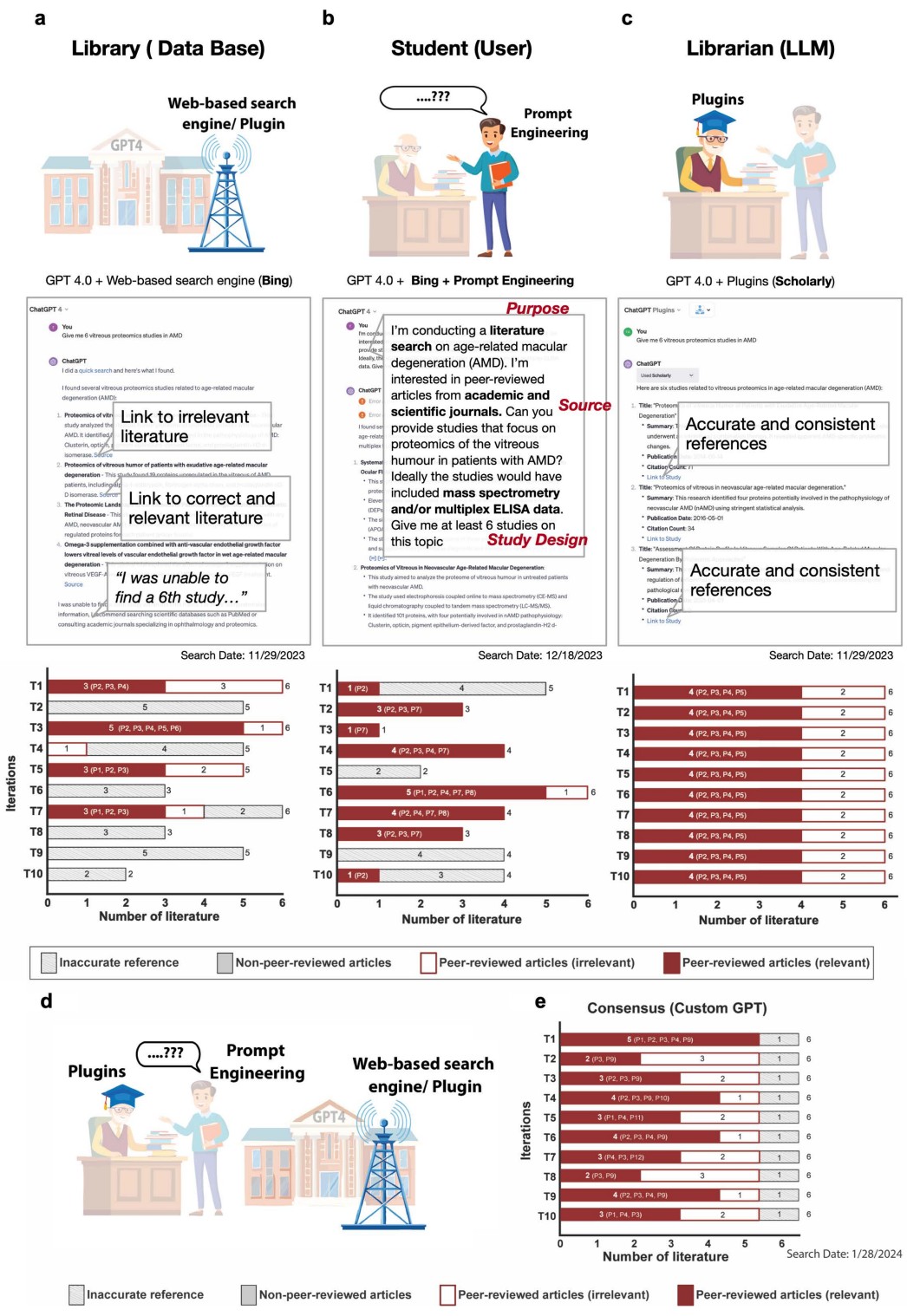

**Fig 2. Can the search for scientific literature be improved by sser-support tools ?** Illustrations in panels a, b, and c were generated using Adobe Illustrator's generative AI features. Relevant peer-reviewed publications retrieved: P1 [34], P2 [35], P3 [36], P4 [37], P5 [38], P6 [39], P7 [40], P8 [41], P9 [42], P10 [43], P11 [44].

On Jan 10th 2024, OpenAI introduced the Custom GPT store, allowing users to create and use customized ChatGPT versions tailored for specific purposes. These customized models can have a specified knowledge base, pre-configured prompts, and additional functionalities, essentially combining the 3 potential ways of augmentations mentioned earlier into one framework [28]. We tested Consensus, one of the popular GPTs used for academic searching [29], and noted that in ten iterations, it identified 3 additional papers that were previously not discovered using other search approaches. This improvement may be due to Consensus's functionality of accessing information specifically from the Semantic Scholar database, using a fine-tuned LLM specifically trained to understand and summarize research articles, and an advanced in-built search algorithm that improves literature search responses [30].

Overall, augmentations significantly improved ChatGPT's ability to return relevant research manuscripts with accurate sources. Yet, it couldn't fully resolve issues with consistency, accuracy, and relevancy. It's important to highlight that ChatGPT's web-browsing function is restricted to open-source information, excluding subscription-based journals or publications. While there are multiple academic search plugins, including "Scholarly", that can be paired with ChatGPT to access a broader range of literature databases, users should remain cognizant of the limitations regarding the coverage and timeliness these sources provide.

## Practical evaluation of ChatGPT-based scientific literature search in various scenarios

**Using ChatGPT for literature searches in information-rich topics.** Augmented ChatGPT proved useful, unlike its basic counterpart, when we focused on the specific scientific findings of the most prevalent patient disease in ophthalmology. We extended our tests to COVID-19, a topic that is of high interest to the public and embodies a high volume of publications, to observe if the abundance of information would affect ChatGPT's response.

We asked ChatGPT to "show all studies that discuss genetic risk factors for Long COVID-19" (**Fig 3A**), testing this five times in both the basic and augmented ChatGPT models. Basic ChatGPT models failed to return any answers and suggested conventional search methods for this task. Interestingly, in all five iterations, web-based ChatGPT-4 identified six articles with clear sources, but these were predominantly news articles rather than peer-reviewed studies (**Fig 3B**). In contrast, the scholarly-augmented version consistently returned the same three or four relevant research articles (**Fig 3D**).

To enhance answer quality, we refined our prompt, specifying the need for peer-reviewed article sources, and retested all four approaches (**Fig 3A**). The web-browsing ChatGPT-4 model showed more research and fewer news articles per iteration (**Fig 3C**), however the refinement had little effect on the responses from both basic ChatGPT and ChatGPT augmented with Scholarly (**Fig 3E**). Our findings indicate that basic ChatGPT models were not particularly effective for literature searches in this context. Although the prompt-engineered, augmented model could return relevant literature, the vast amount of available information appeared to hinder its capability to consistently identify peer-reviewed studies, which are often the primary interest of scientists.

**Using ChatGPT for hypothesis generation.** Literature searches not only assess current literature landscape but also aid in hypothesis generation focusing on novelty, reasonability, and testability. We evaluated if web-based ChatGPT-4 could return relevant literature and contextualize it in the logical frame of our hypothesis. We hypothesized that "creatine intake is associated with hair loss (AC)" based on our rationale that "creatine intake increases DHT (AB)" and "an increase in DHT is associated with hair loss (BC) (**Fig 4A**)." Initiating with the query (AC): "Provide relevant peer-reviewed studies that are related to our hypothesis: creatine intake can lead to hair loss." The responses were inconsistent and returned only one relevant research article either showed only AB correlation, or a review paper discussing potential side effects of creatine intake in five iterations (**Fig 4B**). Recently, it was suggested that ChatGPT may generate better-quality responses when it is prompted to reason through steps by splitting complex tasks into simpler, intermediate tasks. This prompting method, termed "Chain-of-Thought Prompting" can enable the LLM to focus on fewer variables at a time with more computation, resulting in lower error rates and improving the LLM's reasoning ability (https://platform.openai.com/docs/guides/prompt-engineering) [31].

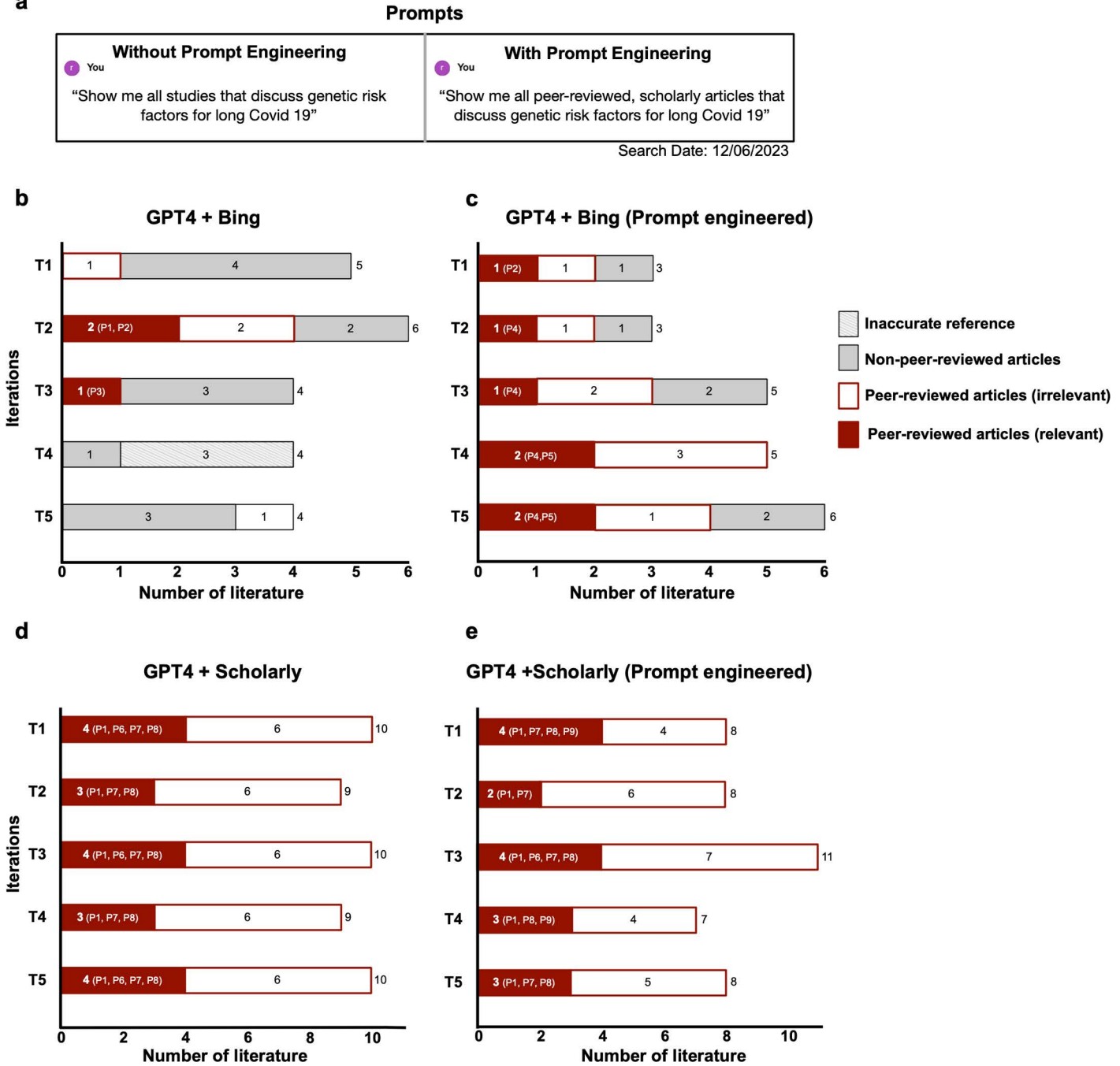

**Fig 3. Literature search results with augmented ChatGPT on high-interest topic with abundance of information (COVID- 19).** Relevant peer-reviewed publications retrieved: P1 [34], P2 [35], P3 [36], P4 [37], P5 [38], P6 [39], P7 [40], P8 [41].

Therefore, we prompt engineered to have ChatGPT follow the human logical flow of generating a hypothesis, by querying separately in an AB, BC manner (**Fig 4C**). This strategy generally improved the number of relevant peer-reviewed articles retrieved. There was better contextualization of literature in our logical framework of the hypothesis, also including future directions proposals. Yet, it's important to note that a prompt engineered approach still showed inconsistency and

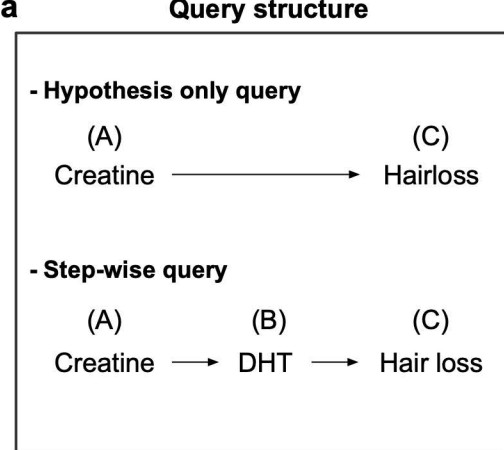

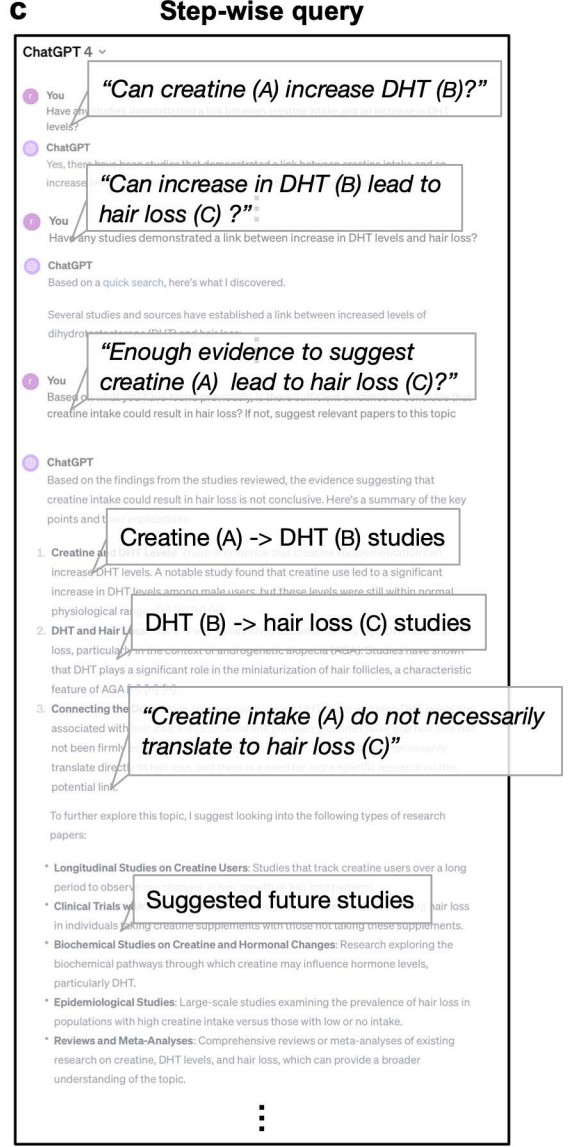

**Fig 4. Can augmented ChatGPT assist researchers in hypothesis generation.**

variability in the quality of responses. As such, employing a step-by-step inquiry approach, mirroring human logical reasoning, might optimize the utility of ChatGPT in hypothesis generation.

**Using ChatGPT for searching clinical practice guidelines.** Decision-making in clinical practice is typically guided by established standards of care. However, clinicians often face cases or scenarios where these standards are not well-established due to contradictory or inconclusive evidence. For instance, semaglutide (commonly known as Ozempic), a drug commonly used for the treatment of type 2 diabetes, has been shown to impact gastric emptying which may be a risk for operations that require general anesthesia [32]. Although the American Society of Anaesthesiologists has published a consensus-based guideline, it recognized the "limited" evidence for establishing a preoperative fasting standards for patients taking Ozempic [33]. In clinical practice, where guidelines require accuracy and conclusiveness for patient

care, the precision of information is crucial. To assess ChatGPT's utility in clinical setting, we asked it to "show me all studies that discuss preoperative fasting guidelines for Ozempic". We hoped that it not only can identify previous relevant studies and the most up-to-date guidelines but also acknowledge the lack of evidence. The basic ChatGPT models failed to identify relevant guidelines, and even the augmented ChatGPT with Scholarly, despite its prevoiusly promising performance, failed to find any pertinent studies related to our query in all five iterations. In contrast, ChatGPT-4 with web-browsing functionality identified the guidelines and but only noted the lack of evidence in two of the five iterations. These results indicate that, as of now, all versions of ChatGPT may not be sufficiently reliable for literature searches specifically aimed at finding clinical guidelines.

## Conclusion

This manuscript provides a snapshot view of LLM's utility in literature searching at the time of testing from November 2023 to early 2024. As of March 2025, LLM and generative AI's utility in literature search is no longer solely available in the form of conversational AI such as ChatGPT. Many of the search engines referred to "conventional" in this manuscript such as Google have already incorporated AI features to enhance literature searches available to users such as AI-generated summaries in Google responses. This manuscript evaluated ChatGPT's utility by conducting search scenarios in five to ten independent iterations. As LLMs quickly advance, we suggest future studies to increase the number of iterations for a more holistic understanding of the limitations of ChatGPT across different search scenarios.

At first glance, AI holds great possibilities with assisting scientific researchers on time-consuming and mundane tasks such as literature searches. However, the inconsistent accuracy underscores the need for careful human oversight. Despite this, the conversational AI tools are advancing rapidly with LLMs continuing to be optimized with plugins adding additional functions. We envision a time when AI becomes a strong and reliable ally in streamlining and reshaping scientific research practices. Yet the question persists, is now truly the right time for the scientific community to fully embrace the utilization of such tools? As we navigate the balance between AI's potential benefits and the imperative for rigorous scientific integrity, this question remains central to the ongoing discourse on the role of AI in research (this concluding paragraph was suggested by ChatGPT based on this article with several iterations of human refinement).

## Methods

### Artificial Intelligence and web-based search engines

Queries with web-based search engines (Google, PubMed, and Google Scholar) were conducted on November 28, 2023. For evaluating basic functions of ChatGPT, ChatGPT 3.5 and ChatGPT Classic were employed on November 28, 2023. Access to these models were facilitated through the drop-down menu in the ChatGPT user interface. To verify the activation of the web-browsing feature of ChatGPT 4, we looked for responses that explicitly referenced web sources. The Scholarly plugin was accessed via the "ChatGPT Plugin" option from the dropdown menu in the ChatGPT interface. Additionally, Consensus GPT was accessed through the ChatGPT GPT store.

### Evaluation on consistency, relevancy and accuracy

Each query was tested in 5–10 independent iterations, where each iteration was conducted as a new ChatGPT session to prevent context retention bias. To assess for accuracy, we manually verified whether the manuscript titles in ChatGPT responses match actual publications that can be identified in Google/Google Scholar/PubMed repository. For cases where a hyperlink is provided in the response (e.g., from web-browsing GPT, plugins or custom GPT), we added an additional criterion to check if the provided link correctly directed to the article cited in its response. Note that an "accurate reference" in the scientific context also includes additional criteria such as correct authorship, publication date, and journal name. We did not include these in our criteria as we focused on assessing ChatGPT's utility in real-world scenarios for identifying literature.

A response can only be used to assess for relevancy if it provided accurate reference as defined above. We proofread each individual article that was returned and checked if it contained relevant information pertaining to the query in testing. Responses from all the iterations for a specific query were compared to evaluate consistencies across independent sessions. For queries conducted on search engines (i.e., Google, PubMed, and Google Scholar), we proofread and evaluated the top ten results.

## Supporting information

**S1 File. ChatGPT conversation transcripts.**
(PDF)

## Acknowledgments

We express our appreciation to Julian Wolf and Charles Meno Theodore Deboer at Stanford Ophthalmology for their insights regarding data presentation and evaluations of queries investigated in this manuscript. We thank MaryAnn Mahajan and Joel Andrew Franco at Stanford Ophthalmology for their proofreading of the manuscript.

## Author contributions

**Conceptualization:** Rui Yip, Young Joo Sun, Alexander Bassuk, Vinit B. Mahajan.

**Data curation:** Rui Yip.

**Formal analysis:** Rui Yip.

**Funding acquisition:** Alexander Bassuk, Vinit B. Mahajan.

**Investigation:** Rui Yip, Young Joo Sun, Vinit B. Mahajan.

**Methodology:** Rui Yip, Young Joo Sun.

**Resources:** Vinit B. Mahajan.

**Supervision:** Young Joo Sun, Vinit B. Mahajan.

**Visualization:** Rui Yip, Young Joo Sun.

**Writing – original draft:** Rui Yip, Young Joo Sun.

**Writing – review & editing:** Rui Yip, Young Joo Sun, Vinit B. Mahajan.

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
