## [Decision Letter · Decision Letter 0]

21 Feb 2025

PDIG-D-24-00172Artificial Intelligence’s Contribution to Biomedical Literature Search: Revolutionizing or Complicating?PLOS Digital Health Dear Dr. Mahajan, Thank you for submitting your manuscript to PLOS Digital Health. After careful consideration, we feel that it has merit but does not fully meet PLOS Digital Health's publication criteria as it currently stands. Therefore, we invite you to submit a revised version of the manuscript that addresses the points raised during the review process. Please submit your revised manuscript within 30 days Mar 23 2025 11:59PM. If you will need more time than this to complete your revisions, please reply to this message or contact the journal office at digitalhealth@plos.org. Please include the following items when submitting your revised manuscript: * A rebuttal letter that responds to each point raised by the editor and reviewer(s). You should upload this letter as a separate file labeled 'Response to Reviewers '. This file does not need to include responses to any formatting updates and technical items listed in the 'Journal Requirements' section below.* A marked-up copy of your manuscript that highlights changes made to the original version. You should upload this as a separate file labeled 'Revised Manuscript with Track Changes '.* An unmarked version of your revised paper without tracked changes. You should upload this as a separate file labeled 'Manuscript '. If you would like to make changes to your financial disclosure, competing interests statement, or data availability statement, please make these updates within the submission form at the time of resubmission. Guidelines for resubmitting your figure files are available below the reviewer comments at the end of this letter. We look forward to receiving your revised manuscript. Kind regards, Luis Filipe Nakayama, M.D.Academic EditorPLOS Digital Health Luis NakayamaAcademic EditorPLOS Digital Health Leo Anthony CeliEditor-in-ChiefPLOS Digital Healthorcid.org/0000-0001-6712-6626  **Journal Requirements:** 1. We have amended your Competing Interest statement to comply with journal style. We kindly ask that you double check the statement and let us know if anything is incorrect.  2. Please provide an Author Summary. This should appear in your manuscript between the Abstract (if applicable) and the Introduction, and should be 150–200 words long. The aim should be to make your findings accessible to a wide audience that includes both scientists and non-scientists. Sample summaries can be found on our website under Submission Guidelines:  https://journals.plos.org/digitalhealth/s/submission-guidelines#loc-parts-of-a-submission  3. In the online submission form, you indicated that “Any additional information required to reanalyze the data reported in this paper is available from the lead contact upon request.” All PLOS journals now require all data underlying the findings described in their manuscript to be freely available to other researchers, either 1. In a public repository, 2. Within the manuscript itself, or 3. Uploaded as supplementary information. This policy applies to all data except where public deposition would breach compliance with the protocol approved by your research ethics board. If your data cannot be made publicly available for ethical or legal reasons (e.g., public availability would compromise patient privacy), please explain your reasons by return email and your exemption request will be escalated to the editor for approval. Your exemption request will be handled independently and will not hold up the peer review process, but will need to be resolved should your manuscript be accepted for publication. One of the Editorial team will then be in touch if there are any issues.  4. Some material included in your submission may be copyrighted. According to PLOS’s copyright policy, authors who use figures or other material (e.g., graphics, clipart, maps) from another author or copyright holder must demonstrate or obtain permission to publish this material under the Creative Commons Attribution 4.0 International (CC BY 4.0) License used by PLOS journals. Please closely review the details of PLOS’s copyright requirements here: PLOS Licenses and Copyright. If you need to request permissions from a copyright holder, you may use PLOS's Copyright Content Permission form. Please respond directly to this email or email the journal office and provide any known details concerning your material's license terms and permissions required for reuse, even if you have not yet obtained copyright permissions or are unsure of your material's copyright compatibility.  Potential Copyright Issues: Figure 2: Please confirm whether you drew the images / clip-art within the figure panels by hand. If you did not draw the images, please provide (a) a link to the source of the images or icons and their license / terms of use; or (b) written permission from the copyright holder to publish the images or icons under our CC-BY 4.0 license. Alternatively, you may replace the images with open source alternatives. See these open source resources you may use to replace images / clip-art:- https://commons.wikimedia.org-
https://openclipart.org/ **Additional Editor Comments (if provided):** Review of “Artificial Intelligence’s Contribution to Biomedical Literature Search: Revolutionizing or Complicating?”. This manuscript addresses an important topic, highlighting a crucial issue regarding the role of LLMs in research. The study is relevant, given the increasing reliance on AI for literature searches in biomedical research.

1) I suggest improving the pubmed search strategy, clarifying the the use of a prompt as a search strategy with an acronym instead of a keyword search.

2) I suggest improving the quality of the figures, as the details are difficult to read in their current form. Clearer visuals would significantly aid reviewers and readers’ comprehension.

3) The Methods section would benefit from reorganization, as some methodological descriptions appear within the Results and Discussion sections.**Reviewers' Comments:** Reviewer's Responses to Questions

**Comments to the Author**

1. Does this manuscript meet PLOS Digital Health’s publication criteria ? Is the manuscript technically sound, and do the data support the conclusions? The manuscript must describe methodologically and ethically rigorous research with conclusions that are appropriately drawn based on the data presented.

Reviewer #1: Yes

Reviewer #2: Yes

2. Has the statistical analysis been performed appropriately and rigorously?

Reviewer #1: No

Reviewer #2: Yes

3. Have the authors made all data underlying the findings in their manuscript fully available (please refer to the Data Availability Statement at the start of the manuscript PDF file)?

Reviewer #1: Yes

Reviewer #2: Yes

4. Is the manuscript presented in an intelligible fashion and written in standard English?

Reviewer #1: Yes

Reviewer #2: Yes

5. Review Comments to the Author

Reviewer #1: The manuscript investigates the effectiveness of ChatGPT, alongside traditional literature search methods such as PubMed and Google, in assisting biomedical researchers and clinicians with literature searches. The study evaluates ChatGPT in various scenarios, testing its basic functions and exploring enhancements through user-support tools (plugins, prompt engineering, web-browsing). The authors highlight both the limitations and potential of ChatGPT in retrieving consistent, relevant, and accurate scientific literature.

while the paper provides a timely and impactful evaluation of ChatGPT's potential to assist researchers with literature searches, offering valuable insights into its current limitations and highlighting the future potential of AI-augmented search tools for scientific research, it also has potential areas for Improvement:

- While the comparison with PubMed and Google is insightful, the manuscript states that no search optimization strategies (e.g., Boolean operators) were used in PubMed to ensure fairness. However, PubMed’s strength lies in its ability to narrow results using advanced search functionalities, not using these functions could understate the potential of conventional methods, mainly when additional settings such as prompt engineering or web search are included in ChatGPT.

- While plugins and prompt engineering are tested, the explanation of how these modifications improved or failed is somewhat superficial. For example, how exactly does “prompt engineering” improve accuracy and relevance? A more granular analysis of the specific improvements introduced by each augmentation would strengthen the conclusions. Additionally, sharing the prompts used could improve the reliability and reproducibility of the experiments.

- Although the manuscript mentions "consistency," "accuracy," and "relevance" as evaluation metrics, these terms are not defined with clear, measurable criteria. Providing concrete definitions or scoring systems for these terms would allow for more rigorous assessment. For instance, how is "accuracy" quantified when ChatGPT provides a reference—what makes a reference “accurate” beyond its correct citation formatting?

- The manuscript limits each search scenario to five or ten iterations, which is a relatively small sample size for assessing the variability in outputs. This is especially important when evaluating AI, as its performance can vary significantly depending on input nuances. Expanding the number of iterations per scenario and including other statistical measures could provide a more robust understanding of ChatGPT's limitations and strengths.

- The quality of the images in the manuscript needs improvement, as they are difficult to review in their current form. Additionally, the accompanying descriptions lack clarity and should provide more detailed explanations to ensure the figures effectively support the findings and are easily interpretable by readers.

Reviewer #2: ChatGPT and Biomedical Literature Search, A Review

One could not think of a timelier and more relevant subject to review and explore than this topic. Especially for a researcher exploring the bioethics of artificial intelligence (AI) and an end user of various forms of generative AI in reading, analyzing and researching medical, surgical and academic texts. More relevant items discussed in this preprint are ophthalmic and diabetes related examples. One can summarize the work steps that the authors followed as steps of comparisons, functions and steps in the field of research in general and the field of research using ChatGPT, as a representation of generative AI. The preprint describes the trend in using generative AI in biomedical research denoting the excitement and the anticipation in adopting its models in the field. Despite the excitement, there are few concerns mentioned in the preprint and its listed references.

The Comparisons:

• Comparing between conventional web-based search methods and generative AI, namely ChatGPT. The conventional search follows a list of few steps to execute the search task, while ChatGPT provides time efficiency through automating the search, extraction and analysis of the information to give immediate response.

• Comparison of basic ChatGPT versions and augmented ChatGPT versions. The basic versions showed limitations in consistency, accuracy and relevancy. The augmented ChatGPT used tools like plugins, web browsing functions, prompt engineering and custom-GPTs.

• Comparison based on scenarios and topics. For example, high interest topics and niche topics (with limited information resources).

• Comparison based on functionalities, like hypothesis generation and newly emerging clinical practice questions.

The Functions:

• The literature search execution by conventional search methodologies, by basic ChatGPT and by augmented ChatGPT.

• The evaluation metrics were the consistency in responses to repeated search queries, the accuracy evaluated by the verification of the references, the authors and other contents of the search, and the relevancy as evaluated by proofreading the “articles returned and checked if it contains relevant information pertaining to the query in testing”

The Tools:

• Basic ChatGPT models: GPT-3.5, ChatGPT Classic.

• Augmented ChatGPT using:

1- web browsing function for real-time data access.

2- plugins specialized in scientific literature research to tailor how the LLM interprets and conveys information with the “Scholarly” plugin.

3- Prompt engineering, which ensures asking the right questions and providing clear extraction.

4- Custom GPT. For example, Consensus GPT for academic research.

• Conventional search engines: Google, PubMed, and Google Scholar

Another way to review this preprint can also be put in another framework is as follows:

• Methodology including description of the conventional search engines and detailed steps of employing ChatGPT for literature research.

• The evaluation and analysis included comparisons between conventional search engines and ChatGPT, quantitative assessment based on the number of iterations and times, and the numbers of results rendered by the various search engines. Qualitative analysis included exploring challenges in different scenarios. For example, high interest topics and niche topics. Functions like hypothesis generation and clinical practice guidelines.

• The preprint also discussed prospects for involving ChatGPT in literature research. These prospects included the rapid advancement and the augmentation of conversational AI, which will come with the two mentioned advantages of time saving and improving accessibility to research resources. Hypothesis generation can also be developed for better utilization of generative AI in research. All the mentioned limitations are prospected to be advantages for ChatGPT in the future.

• Considering human oversight, some reviews discuss it as a disadvantage of ChatGPT and other conversational AI models. This review is with the viewpoint that human oversight is crucial for the role of generative AI now and the future. Other studies presented hybrid approach in literature research, where human experts have more active role in the research to augment ChatGPT in the research process. 1

• Ethical implications include the integrity of research and the potential misuse of ChatGPT and other LLMs in biomedical literature research. Other scientific papers called for setting guidelines for LLM biomedical research deployment. 2 Despite the growing research body about the topic, it seems too early to configure the guidelines and more programed work and comparisons between the research outcomes are needed to discuss the guidelines.

• ChatGPT augmentation strategies improved accuracy of the returned resources, enhanced consistency, gave access to real-time web through real-time web browsing function, enhanced the output quality using prompt engineering as a mode of input modification, different GPTs might render different results, and some can come with more papers than others with the continuously increased iterations.

• Prompt engineering enhances quality and relevance of the responses, but it needs multiple iterations. In addition to the persistence of inconsistency and inaccuracy mentioned in the paper, it can mean the infusion of bias into the research structure and this bias might double fold with the increased iterations. The continuously changing research trends in healthcare also prompt continuous update of the engineered prompts in the deployment of ChatGPT. 3

• ChatGPT is already being tried for integration with the conventional search engines. 4 Conventional search engines including Google, Microsoft Bing and others are already implementing functionalities like ChatGPT to make their search more conversational. So, conventional search engines mentioned in the study are no longer conventional.

References:

1. Temsah, O., Khan, S.A., Chaiah, Y., Senjab, A., Alhasan, K., Jamal, A., Aljamaan, F., Malki, K.H., Halwani, R., Al-Tawfiq, J.A. and Temsah, M.H., 2023. Overview of early ChatGPT’s presence in medical literature: insights from a hybrid literature review by ChatGPT and human experts. Cureus, 15(4).

2. Sallam, M., 2023, March. ChatGPT utility in healthcare education, research, and practice: systematic review on the promising perspectives and valid concerns. In Healthcare (Vol. 11, No. 6, p. 887). MDPI.

3. Abhari, S., Fatahi, S., Saragadam, A., Chumachenko, D. and Morita, P.P., 2024. A Road Map of Prompt Engineering for ChatGPT in Healthcare: A Perspective Study. Studies in Health Technology and Informatics. IOS Press. https://doi. org/10.3233/SHTI240578.

4. Stokel-Walker, C., 2023. AI chatbots are coming to search engines—can you trust the results?.

6. PLOS authors have the option to publish the peer review history of their article (what does this mean? ). If published, this will include your full peer review and any attached files.

**Do you want your identity to be public for this peer review?** For information about this choice, including consent withdrawal, please see our Privacy Policy .

Reviewer #1: **Yes: ** David Restrepo

Reviewer #2: **Yes: ** Yasser Abdullah

**Figure resubmission:** While revising your submission, please upload your figure files to the Preflight Analysis and Conversion Engine (PACE) digital diagnostic tool, https://pacev2.apexcovantage.com/. PACE helps ensure that figures meet PLOS requirements. To use PACE, you must first register as a user. Registration is free. Then, login and navigate to the UPLOAD tab, where you will find detailed instructions on how to use the tool. If you encounter any issues or have any questions when using PACE, please email PLOS at figures@plos.org. Please note that Supporting Information files do not need this step. If there are other versions of figure files still present in your submission file inventory at resubmission, please replace them with the PACE-processed versions.**Reproducibility:** To enhance the reproducibility of your results, we recommend that authors of applicable studies deposit laboratory protocols in protocols.io, where a protocol can be assigned its own identifier (DOI) such that it can be cited independently in the future. Additionally, PLOS ONE offers an option to publish peer-reviewed clinical study protocols. Read more information on sharing protocols at https://plos.org/protocols?utm_medium=editorial-email&utm_source=authorletters&utm_campaign=protocols

---

## [Decision Letter · Decision Letter 1]

3 Apr 2025

Artificial Intelligence’s Contribution to Biomedical Literature Search: Revolutionizing or Complicating?

PDIG-D-24-00172R1

Dear Prof. Mahajan,

We are pleased to inform you that your manuscript 'Artificial Intelligence’s Contribution to Biomedical Literature Search: Revolutionizing or Complicating?' has been provisionally accepted for publication in PLOS Digital Health.

Best regards,

Luis Filipe Nakayama, M.D.

Academic Editor

PLOS Digital Health

**Additional Editor Comments (if provided):**

Thank you for your revisions. The manuscript is now improved and suitable for publication.

**Reviewer Comments (if any, and for reference):**

Reviewer's Responses to Questions

**Comments to the Author**

1. If the authors have adequately addressed your comments raised in a previous round of review and you feel that this manuscript is now acceptable for publication, you may indicate that here to bypass the “Comments to the Author” section, enter your conflict of interest statement in the “Confidential to Editor” section, and submit your "Accept" recommendation.

Reviewer #1: All comments have been addressed

2. Does this manuscript meet PLOS Digital Health’s publication criteria ? Is the manuscript technically sound, and do the data support the conclusions? The manuscript must describe methodologically and ethically rigorous research with conclusions that are appropriately drawn based on the data presented.

Reviewer #1: Yes

3. Has the statistical analysis been performed appropriately and rigorously?

Reviewer #1: Yes

4. Have the authors made all data underlying the findings in their manuscript fully available (please refer to the Data Availability Statement at the start of the manuscript PDF file)?

Reviewer #1: Yes

5. Is the manuscript presented in an intelligible fashion and written in standard English?

Reviewer #1: Yes

6. Review Comments to the Author

Reviewer #1: The authors have comprehensively addressed all the comments. The study robustly illustrates both the limitations and the potential of integrating AI tools into biomedical literature searches. The paper thereby contributes valuable insights into how AI can both contribute to and complicate the research process, emphasizing the need for ongoing refinements and rigorous evaluations as these tools evolve.

7. PLOS authors have the option to publish the peer review history of their article (what does this mean? ). If published, this will include your full peer review and any attached files.

**Do you want your identity to be public for this peer review?** For information about this choice, including consent withdrawal, please see our Privacy Policy .

Reviewer #1: **Yes: ** David Restrepo
